# Autoantibodies against Complement Classical Pathway Components C1q, C1r, C1s and C1-Inh in Patients with Lupus Nephritis

**DOI:** 10.3390/ijms23169281

**Published:** 2022-08-17

**Authors:** Maria Radanova, Vasil Vasilev, Galya Mihaylova, Mariya Kosturkova, Uday Kishore, Lubka Roumenina

**Affiliations:** 1Department of Biochemistry, Molecular Medicine and Nutrigenomics, Medical University of Varna, 9000 Varna, Bulgaria; 2Clinic of Nephrology, University Hospital “Tsaritza Yoanna—ISUL”, 1000 Sofia, Bulgaria; 3Department of Nephrology, Medical University of Sofia, 1000 Sofia, Bulgaria; 4Department of Propaedeutics of Internal Diseases, Medical University of Varna, 9000 Varna, Bulgaria; 5Department of Life Sciences, College of Health, Medicine and Life Sciences, Brunel University, London UB8 3PH, UK; 6Department of Veterinary Medicine, U.A.E. University, Al Ain P.O. Box 1551, United Arab Emirates; 7Centre de Recherche des Cordeliers, Institut National de la Santé et de la Recherche Médicale (INSERM), Sorbonne Universités, Université de Paris Cite, F-75006 Paris, France

**Keywords:** lupus nephritis, complement, anti-complement autoantibodies

## Abstract

Autoantibodies against the complement component C1q (anti-C1q) are among the main biomarkers in lupus nephritis (LN) known to contribute to renal injury. C1q, the recognition subcomponent of the complement classical pathway, forms a heterotetrameric complex with C1r and C1s, and can also associate a central complement regulator and C1 Inhibitor (C1-Inh). However, the frequency and the pathogenic relevance of anti-C1r, anti-C1s and anti-C1-Inh autoantibodies remain poorly studied in LN. In this paper, we screened for anti-C1q, anti-C1r, anti-C1s and anti-C1-Inh autoantibodies and evaluated their association with disease activity and severity in 74 LN patients followed up for 5 years with a total of 266 plasma samples collected. The presence of anti-C1q, anti-C1r, anti-C1s and anti-C1-Inh was assessed by ELISA. IgG was purified by Protein G from antigen-positive plasma and their binding to purified C1q, C1r and C1s was examined by surface plasmon resonance (SPR). The abilities of anti-C1q, anti-C1r and anti-C1s binding IgG on C1 complex formation were analyzed by ELISA. The screening of LN patients’ plasma revealed 14.9% anti-C1q positivity; only 4.2%, 6.9% and 0% were found to be positive for anti-C1r, anti-C1s and anti-C1-Inh, respectively. Significant correlations were found between anti-C1q and anti-dsDNA, and anti-nuclear antibodies, C3 and C4, respectively. High levels of anti-C1q antibodies were significantly associated with renal histologic lesions and correlated with histological activity index. Patients with the most severe disease (A class according to BILAG Renal score) had higher levels of anti-C1q antibodies. Anti-C1r and anti-C1s antibodies did not correlate with the clinical characteristics of the LN patients, did not interfere with the C1 complex formation, and were not measurable via SPR. In conclusion, the presence of anti-C1q, but not anti-C1s or anti-C1r, autoantibodies contribute to the autoimmune pathology and the severity of LN.

## 1. Introduction

Systemic lupus erythematosus (SLE) is a prototype autoimmune disease with complex pathology; one of its common and severe manifestations is lupus nephritis (LN) [1,2]. The complement classical pathway is well known to have an association with the disease pathogenesis [3]. It is activated when antigen–antibody complexes bind to the C1 complex, leading to the proteolytic cleavage of C4 and C2 complement proteins to yield the C3 convertase of the classical pathway. C1 complex is composed of C1q and two serine proteases, C1r and C1s (C1q + C1r2 + C1s2) [4,5]. Autoantibodies against complement components and regulators result in acquired functional deficiencies related to the complement cascade in SLE and LN [6]. One of the most pathologically important anti-complement autoantibodies in SLE and LN are those targeting C1q. Anti-C1q antibodies are one of the biomarkers used for the evaluation of lupus nephropathy. High anti-C1q antibody levels are present in the sera of approximately 20% to 50% of the SLE patients [7,8,9]. The positive predictive value of anti-C1q antibodies for the development of LN has been estimated to be about 58%, while its negative predicting value for LN ranges between 91% and 100% [10,11,12,13]. Raised titers of anti-C1q antibodies have a predictive value for the exacerbation and recurrence of LN; in addition, they are associated with proliferative nephritis forms [6,7,9,14,15,16,17,18,19]. However, data on the presence of autoantibodies against C1r in SLE and LN patients are scarce; autoantibodies against C1s have been reported in 7/15 patients with SLE and LN [20]. C1s appears to show nearly four-fold higher proteolytic activity in the presence of anti-C1s antibody, thus contributing to the amplification of the complement-dependent cellular lysis, and availability of autoantigens, and hence, the possible development of autoimmunity [20]. In the current study, we have examined the frequency and the pathogenic relevance of anti-C1q, anti-C1r, anti-C1s and anti-C1-Inh autoantibodies in LN patients confirmed via biopsy, in order to ascertain if autoantibodies against all subcomponents of the C1 complex have any prognostic value in the disease.

## 2. Results

### 2.1. Autoantibodies Recognizing the Components of the C1 Complex Are Present in the Plasma of LN Patients

An ELISA-based experiment was set up to determine the presence of anti-C1r, C1s and anti-C1-Inh autoantibodies in LN patients’ plasma for comparison with corresponding levels of anti-C1q autoantibodies. In total, 11 out of 74 patients (14.86%) were seropositive for anti-C1q autoantibodies (Figure 1A); all these patients had the active disease. Only 3 out of 72 tested patients (4.16%) were seropositive for anti-C1r autoantibodies; 2 of them had the active disease (66.67%, 2/3) (Figure 1B). 5 out of the 72 tested patients (6.94%) were seropositive for anti-C1s autoantibodies (Figure 1C), and 2 had the active disease (40.00%, 2/5). The binding of anti-C1r autoantibodies to C1r and that of anti-C1s autoantibodies to C1s was dose-dependent (Figure 1D,E). No patient was found positive for anti-C1 Inhibitor autoantibodies. Among all seropositive plasma for C1 antibodies, there was no plasma simultaneously positive for anti-C1q, anti-C1r and anti-C1s autoantibodies; one patient had both anti-C1q and anti-C1s autoantibodies and two patients had anti-C1q as well as anti-C1r autoantibodies. Five patients were found to be seropositive for anti-C1r as well as anti-C1s autoantibodies. Moreover, during the follow up analysis of the patients, we found that four patients became positive for anti-C1q and anti-C1r, seven had anti-C1q and anti-C1s, and seven had both anti-C1r and anti-C1s autoantibodies. Only one patient became seropositive for the three proteins, i.e., C1q, C1r and C1s, during the follow-up study.

Among the healthy volunteers, high levels of anti-C1q autoantibodies were detected in five individuals (6.17%, 5/81); however, there were no healthy subjects seropositive for anti-C1r and anti-C1s autoantibodies (Figure 1A–C, F–H).

We also examined whether some of the patients became seropositive for anti-C1q, anti-C1r or anti-C1s antibodies during the follow-up period. For all 74 patients, 266 samples were available. We recalculated the number of patients, seropositive for anti-C1q, anti-C1r or anti-C1s antibodies, considering the sample with the highest level of anti-C1q, anti-C1r and anti-C1s antibodies for each patient, respectively. Thus, 20/74 (27.03%) samples were found seropositive for anti-C1q, 16/72 (22.22%) for anti-C1r, and 20/72 (27.78%) for anti-C1s antibodies. Significant differences between the frequencies of anti-C1q, anti-C1r and ant-C1s autoantibodies in LN patients and healthy volunteers were found (*p* = 0.0002, *p* = 0.0003 and *p* < 0.0001, respectively) (Figure 1F–H).

The autoantibody levels varied overtime for the patients that were positive for anti-C1q, anti-C1r and anti-C1s antibodies. Three patients maintained elevated anti-C1q levels over time (LN7, LN11 and LN13). For five patients that were positive for anti-C1q autoantibodies (LN2, LN4, LN12, LN14 and LN6), the antibody levels declined to normal and remained low throughout the study period (Figure 2A). The remaining four patients showed various titers returning to normal levels at given time points. In patients with anti-C1r and anti-C1s antibodies, the autoantibody levels showed variation over time (Figure 2B,C).

### 2.2. IgG Purified from LN Patients’ Plasma Show Binding Kinetics via SPR for C1q but Not for C1r and C1s

The anti-C1q positivity of selected patients was validated by SPR (Figure 3A). Four IgG samples from healthy volunteers showed no specific binding (Figure 3B). A set of patients, who were negative by ELISA, remained negative by SPR (Figure 3C). Nevertheless, when an extended analysis was performed with more IgG from LN patients, positive or negative for anti-C1q by ELISA, a discrepancy was observed between the two methods. Some low titer anti-C1q IgG, detected by ELISA, were negative in SPR analysis and vice versa (data not shown).

To find out whether the anti-C1r and anti-C1s autoantibodies did have pathophysiological relevance, we purified IgG from patients’ plasma and attempted to measure their binding affinity for C1r and C1s by SPR. Multiple attempts in different buffers (PBS or HEPES) and concentrations (ranging from 500 µg/mL to 1 µg/mL) failed to reveal a significant binding to C1r as well as C1s (data not shown).

### 2.3. IgG Antibodies from Anti-C1r- and Anti-C1s-Positive Plasma Do Not Affect C1 Complex Formation as Opposed to Anti-C1q Autoantibodies

C1r and C1s are produced by the liver, while C1q is produced primarily by adherent monocytes, tissue macrophages and immature dendritic cells, suggesting that at least a proportion of the C1 complexes is formed in the circulation [21,22,23]. Upon C1 complex formation, large parts of C1r and C1s are hidden within the cone of C1q collagen stalks. To evaluate whether the anti-C1q, anti-C1r and anti-C1s autoantibodies could interfere with their ability to associate, a functional test for C1 complex formation was set up. In this assay, a capture rabbit anti-C1q was used and the presence of the C1 complex was revealed by goat anti-C1s antibodies. The pre-incubation of C1r and C1s with IgG purified from sero-positive and negative patients did not alter the C1-forming capacity (Figure 3D). IgG from patients positive for anti-C1r or anti-C1s antibodies did not influence C1 complex formation. In contrast, the two tested anti-C1q positive IgG induced a dose-dependent decrease in the C1 complex formation.

### 2.4. Association between Anti-C1q, Anti-C1r and Anti-C1s Autoantibodies and LN Severity

The presence of anti-C1q antibodies associated with the higher levels of proteinuria (*p* = 0.023) and the presence of active urinary sediment (*p* = 0.027), but these were not valid for cases with anti-C1r and anti-C1s autoantibodies. Statistically significant but weak inverse correlations between levels of anti-C1q antibodies and plasma C3 and C4 concentrations in patients (for C3 r = −0.213, *p* = 0.0006, Figure 4A and for C4 r = −0.216, *p* = 0.0006, Figure 4B) were observed. A moderately positive correlation between anti-C1q and anti-dsDNA levels was observed (r = 0.453, *p* < 0.0001; Figure 4C). No correlation was found between anti-nuclear antibodies (ANA) and anti-C1q levels in the follow-up plasma (Figure 4D). In addition, no significant correlation between anti-C1r antibodies and the investigated immunological markers was found in the follow-up study (Figure 4E–H). Weak but significant correlations were found for anti-C1s levels and C3 (r = −0.157, *p* = 0.012, Figure 4I), anti-dsDNA (r = 0.150, *p* = 0.025, Figure 4K) levels and ANA titers (r = 0.162, *p* = 0.016, Figure 4L) in the follow-up study. Moreover, patients who were positive for anti-C1s autoantibodies had higher anti-dsDNA antibodies (*p* = 0.018) and ANA (*p* = 0.001), than those negative for anti-C1s antibodies.

### 2.5. Levels of Anti-C1q, Anti-C1r and Anti-C1s Antibodies and LN Activity

Patients with active LN at the time of the first sample collection were more frequently found to be positive for anti-C1q autoantibodies (11/26; 42.31%) than those with partial remission (2/24; 8.33%) and complete remission (1/24; 4.17%). The anti-C1q autoantibodies positively correlated with BILAG Renal score (r = 0.344, *p* = 0.003, Figure 5A). Patients with the most severe disease (A class according to BILAG Renal score) had higher levels of anti-C1q antibodies. Anti-C1q autoantibodies were found more frequently in the patients with severe LN, category A, BILAG Renal score (8/23, 34.78%), in comparison to patients with milder disease, categories B, C or D, BILAG Renal score (3/51, 5.88%), (*p* = 0.002, Figure 5B). ROC analysis also indicated that anti-C1q autoantibodies were a good discriminatory marker (AUC = 0.728, *p* = 0.002, Figure 5C) for LN activity with 50% sensitivity and 74.2% specificity. Neither anti-C1r nor anti-C1s antibodies were found to associate with higher BILAG Renal score category (data not shown).

### 2.6. Presence of Anti-C1q Autoantibodies Associates with Histological Features of LN

The presence of anti-C1q autoantibodies in LN patients at the time of first sample collection was associated with proliferative LN, specifically Class IV (Figure 5D). Levels of anti-C1q antibodies were higher in the Class IV group compared to the Class III (*p* = 0.049) or Class II groups (*p* = 0.010, Figure 5D). 10 out of 13 patients seropositive for anti-C1q antibodies had Class IV LN. 43 out of 58 patients who were negative for anti-C1q antibodies were distributed among Classes I, II, III, V and VI. The presence of anti-C1q antibodies associated strongly with Class IV LN, with 40.0% sensitivity, 93.5% specificity, 76.9% positive predictive value, 74.1% negative predictive value, relative risk—2.97, and OR = 9.56, 95% CI: 2.31–39.45, *p* = 0.0018. When the samples of patients with Class IV only were stratified according to their anti-C1q status, the anti-C1q positive samples had significantly elevated anti-dsDNA (*p* = 0.013), lower C3 (*p* = 0.006) and C4 (*p* = 0.02), and a tendency to increased proteinuria (*p* = 0.062). Moreover, anti-C1q positive Class IV patients had more frequently A class according to BILAG Renal score, contrary to the anti-C1q negative ones (*p* = 0.0286). No difference was found for the eGFR, the index of activity, the active urinary sediment, ANA, the age or the frequency of anti-C1r or anti-C1s.

C1r and C1s autoantibodies did not show significant association with any histological class of LN (data not shown).

Only 27 patients were selected for investigation of the relationship between anti-C1q levels and histological activity and/or chronicity of LN. These were the patients for whom the period between blood sampling and renal biopsy was up to 12 months. Comparative analysis between anti-C1q levels in the groups with and without histological hallmarks of LN activity and chronicity were made (Table 1). Elevated levels of anti-C1q antibodies significantly associated with renal histologic lesions, such as sub-endothelial immune deposits-type “Wire loop” (*p* = 0.019), cellular crescents (*p* = 0.012) and fibrous crescents (*p* = 0.037) (Table 1). A statistically significant positive correlation between levels of anti-C1q and histological activity index was also found (r = 0.432, *p* = 0.025, Figure 5E), but not with histological chronicity index (Figure 5F).

## 3. Discussion

The complement classical pathway is a key driver of the renal injury in LN. Immune complexes trigger the activation of the complement cascade through the classical pathway leading to tissue injury. Sub-endothelial and sub-epithelial deposits of Ig and complement classical pathway components, such as C4 and C1q, are always found in LN patients. Furthermore, the finding IgM, IgG and IgA together with C1q, C3 and C4 in kidney biopsy tissue, often referred as “full house immunofluorescence”, is typical for these patients and is rarely seen in non-lupus renal disease [24]. In this paper, we show that all three components of the C1 complex are targets of autoantibodies in LN. The autoantibodies against C1q have strong correlation with the disease severity and immunological activity, contrary to those against C1r and C1s, which are rarer and show no correlation. Patients with LN usually have autoantibodies against C1q [8,17,25]. Depending on the cohort and geographical origin, the positivity reaches ~97% [15,26], suggesting that LN and anti-C1q autoantibodies are inseparable [14,15,27,28]. Nevertheless, the frequency of anti-C1q in our Bulgarian cohort of LN patients was only 20/74 (27%), which is consistent with earlier studies that reported a frequency of 23% (9/38) [29,30]. The low frequency of active LN in this cohort cannot explain the low anti-C1q antibody frequency, because even among the active LN patients, only 11/26 (42%) (or 8/23 (34.78%) of BILAG A) were found to be anti-C1q-positive. Our protocol for the detection of anti-C1q was ELISA-based [31,32] and carried out in high ionic strength conditions to prevent false-positive results by C1q-containing immune complexes. We also used purified IgG from a set of positive and negative patients and analyzed their binding to immobilized C1q on a SPR chip. For the samples with high titers, binding was detected by both methods, ELSA as well as SPR. However, SPR analysis missed some of the samples with low titer IgG but detected binding in a few samples that were found to be negative by ELISA. The discrepancy between ELISA and SPR found for some samples could be related to the fact that anti-C1q bind weakly (if any) to soluble C1q or to C1q within the C1 [33,34,35] and a conformational change, induced by the tissue/ELISA plate is necessary to reveal the cryptic immunodominant neoepitope(s), such as the so called “A08” epitope [36,37]. These cryptic epitopes may not be correctly exposed upon C1q immobilization to the biosensor chip and the weak titer anti-C1q may not be able to bind them in the cases when the loss of reactivity was detected. On the contrary, other cryptic epitopes may be exposed upon SPR type of immobilization (which is covalent linking via the Lys residues to a flexible dextran matrix, creating a semi-solid, semi-fluid phase model) revealing novel, previously undetected ant-C1q antibodies, missed out by the standard ELISA method. These antibodies need to be further investigated for their functional and clinical relevance. Moreover, immobilized C1 complex could be tested to reveal whether the anti-C1q, anti-C1r, anti-C1s or even the negative IgG samples could reveal some binding due to the exposure of cryptic neoepitopes. Nevertheless, previous finding and our present results suggest that the anti-C1q bind preferentially to the free C1q [38] and even can prevent/dissociate the C1 complex formation. This is in line with the finding that anti-C1q autoantibodies perturb the function of free C1q in the clearance of apoptotic cells [39] and trigger a pro-inflammatory phenotype in macrophages, reversing the anti-inflammatory effects of immobilized C1q alone [40]. Moreover, C1q/anti-C1q-primed monocytes induce pathological T-cell activation via direct CD40–CD154 interaction [41]. Altogether, these results suggest that free and not complexed C1q is the main target of the anti-C1q autoantibodies in LN.

Higher levels of anti-C1q autoantibodies for clinical activity of LN are associated with proteinuria [26,42,43], active urinary sediment [26], lower plasma levels of C4 and C3 [43,44] and pathological levels of anti-dsDNA [43,44]. These associations are consistent with the relations of anti-C1q levels with BILAG Renal score, a clinical tool for the complex evaluation of LN activity, and the determination of the treatment indications. We confirmed the association of anti-C1q with active disease [9,15,27]. The highest levels of anti-C1q were found in patients with IV class LN. Patients with the histological characteristics of active LN (“wire loop” glomerular sub-endothelial deposits and cellular crescents) had significantly higher anti-C1q levels, which also overlapped with other histological signs of LN activity (fibrinoid necrosis/karyorrhexis and interstitial inflammation). These associations are consistent with the significant relation of anti-C1q levels and histological activity index (without establishing such relation of anti-C1q levels with histological chronicity index) as reported previously [45]. Thus, anti-C1q autoantibodies as a non-invasive biomarker offer a promising target for the evaluation of histological LN activity. Such a significant association was not found for anti-C1r and anti-C1s in the current study.

The mechanism of development of an autoimmune response against C1q in LN patients is not well understood. C1q exists primarily as a C1 complex in the circulation. It still remains unclear whether the autoantigen is C1q alone or in the context of the C1 complex. The C1 complex binds to immune complexes and apoptotic cells and activates the classical pathway to bring about opsonization with C3b, which facilitates their clearance [46,47,48]. The dysregulation of the immune response should precede the generation of anti-C1q autoantibodies since they are rare in SLE and their appearance is related to the progression of the renal manifestation [49,50,51]. Indeed, the anti-C1q titers are the highest in the class IV LN, the development of which is frequently preceded by a period of SLE without renal manifestations [44,52,53]. Therefore, the pathological accumulation of immune complexes and dying cells will result in enhanced opsonization with C1 and C3b. Moreover, anti-C1q autoantibodies alone are not able to induce glomerular injury. Anti-C1q autoantibodies deposit in glomeruli but are only pathogenic in combination with glomerular C1q-containing immune complexes [54], which is in line with the observation that anti-C1q autoantibodies are present in other conditions, such as hypocomplementaemic urticarial vasculitis, as well as in healthy individuals, where they are not related to renal pathology. In a murine model, Trouw et al. (2004) demonstrated that glomerular inflammation is caused by anti-C1q antibody only in mice pretreated with C1q-fixing anti-glomerular basement membrane (anti-GBM) antibodies, as a model for glomerular immune complex disease. Thus, the pathological role of anti-C1q antibodies is exerted only when they react with their autoantigen in the context of immune complexes. It is likely that the pathologically relevant autoantigen could be the complex of C1q with C1r and C1s, with or without C1-Inh.

The presence of anti-C1s and anti-C1-Inh autoantibodies has been described in SLE patients, but the cohort studies are scarce [20,55]. Anti-C1s autoantibodies, previously reported in seven SLE patients, show an overt functional activity. They have been shown to stimulate the conversion of C1s pro-enzyme to an active enzyme in the absence of C1q and enhance C4 cleavage in vitro. Nevertheless, no clinical association of these antibodies was reported [20]. In this paper, we found a low frequency of anti-C1r and anti-C1s, and negligible existence of anti-C1-Inh antibodies, at the time of first sample collection, and a higher prevalence of anti-C1r and anti-C1s antibodies when the highest titer samples were used. Although we found patients seropositive for anti-C1s as well as C1r antibodies, their presence and titers correlated poorly with any clinical or immunological parameter. Therefore, they are likely a product of the dysregulated immune response against isolated serine proteases outside the context of C1-complex, independent of the presence of C1q and do not act as pathological factors. C1s and C1r are locally expressed in tissues, including the kidney by cells, which do not express C1q [56,57]. C1s has functions outside the complement cascade as well, acting as an extracellular matrix-degrading enzyme and an enzyme, cleaving nuclear antigens [58,59,60,61,62,63,64,65]. Further studies are needed to unravel the potential functional relevance of the anti-C1s as potentiators of the C4 and/or C2 cleavage and factors affecting the non-canonical functions of C1s, independent of the C1 complex. Indeed, the C1 complex is labile and after the action of C1-Inh, C1s is inactivated and the C1r-C1s-C1Inh complex dissociates from C1, leaving isolated C1q [4]. Therefore, apart from the idea that antibodies target the complement components, bound within the C1 complex, a reaction between antibodies and free from the complex antigen can take place.

## 4. Materials and Methods

### 4.1. LN Patient Cohort

This study involved 74 SLE patients, clinically diagnosed following the criteria of the American College of Rheumatology (ACR). This was a single-center study, based at the Nephrology Clinics of University Hospital “Tsaritza Ioanna−ISUL”, Medical University of Sofia.

BILAG (The British Isles Lupus Assessment Group) Renal score was used to define LN activity [66]. It is an index for measuring clinical disease activity, which assigns patients to different categories, based on the physicians’ intention to treat. The patients were classified into four BILAG categories: 23 patients (31.08%) fell into category A LN (active disease, requiring disease-modifying treatment), 24 patients (32.43%) into category B LN (disease, less active than A, requiring symptomatic treatment), 8 patients (10.81%) into category C LN (stable mild disease), and 19 patients (25.68%) into category D LN (inactive disease).

During the follow-up period, which stretched over 5 years (2011–2016), 52 patients were examined more than once. This study was carried out using 266 plasma samples collected from 74 patients. The cohort included 59 female (79.7%) and 15 male (20.3%) patients, with a median age of 45.4 ± 14.9 years (from 19 to 87) and median LN duration of 10.3 ± 9.5 years (from 0.02 to 41).

Patients with LN, confirmed via renal biopsy, were distributed according to the LN classification of the International Society of Nephrology (ISN) and the Renal Pathology Society (RPS) [67]: 4 patients (5.63%) had LN Class I, 23 (32.39%) had LN Class II, 7 (9.86%) had LN Class III, 25 (35.21%) had LN Class IV, 11 (15.50%) had LN Class V, and 1 (1.41%) had LN Class VI. Only biopsies performed less than 12 months from the sampling for autoantibodies were included in the analysis for correlation with the histological findings.

The positivity for anti-nuclear antibodies (ANA) was determined via indirect immunofluorescence; anti-dsDNA antibody levels were measured by ELISA (and expressed in U/mL). Elevated ANA titers (above 1:80) were found in 50 (69.4%) patients, while elevated levels of anti-dsDNA were found in 31 (40.8%) of patients.

The plasma concentration of the complement components, C4 and C3, was measured by immunodiffusion. The normal values for C3 ranged between 0.75 and 1.65 g/L, and that for C4 varied between 0.20 and 0.65 g/L. C3 hypocomplementemia was detected in 14 patients (19.7%, 14/66), C4 hypocomplementemia in 28 patients (39.4%, 28/71), and concomitant C3 and C4 hypocomplementemia was detected in 13 patients (19.8%).

The control group for this study included 81 healthy individuals, who were age and sex matched to the LN patients. All healthy volunteers were without autoimmune and infectious inflammatory diseases, with normal kidney, liver and hematopoietic functions.

The study was approved by the Ethics Review Boards of Medical University of Varna (protocol No.62/04 May 2017). Each participant signed an informed consent for the enrolment and the study was conducted following the guidelines of the Declaration of Helsinki.

### 4.2. ELISA for Detecting Anti-Complement Autoantibodies

A total of 20 µg/mL of each tested antigen, human C1q, C1r, C1s and C1-Inh (Complement Technology, Ins), were adsorbed on microtiter plates (Greiner Bio-One, Kremsmünster, Austria) in sodium carbonate/bicarbonate buffer (35 mM NaHCO_3_, 15 mM Na_2_CO_3_, pH 9.6) overnight at 4 °C. Each plate was blocked with 1% *w*/*v* Bovine Serum Albumin (BSA) in PBS for 1 h at 37 °C and washed three times with PBS containing 0.05% *v*/*v* Tween-20 (PBST). Plasma samples from each patient were diluted 1/100 in PBST. The ELISA for detecting anti-C1q antibodies involved diluting plasma in PBS/750 mM NaCl-0.1% Tween 20 in order to prevent C1q-IgG complex formation and allow only specific antigen–antibody binding. After washing, horseradish peroxidase (HRP)-conjugated anti-human IgG secondary antibodies (Southern Biotech, Birmingham, AL, USA) were applied at 1/1000 dilution in PBST. Following three rounds of washing with PBST, the color was developed using 0.5 mg/mL o-phenylenediamine (OPD) (ThermoFisher Scientific, Illkirch-Graffenstaden, France) and the signal was detected via optical density (OD) measured at 490 nm.

An evaluation of the dose–response of the binding of the autoantibodies from the pre-defined positive plasma samples was performed by applying serially diluted plasmas starting from 1/50 dilution on coated and blocked microtiter wells. A plasma sample was determined as positive for a given autoantibody if its optical density exceeded the average of the optical density of the samples of the healthy volunteers ±3 standard deviations.

### 4.3. IgG Purification

Plasma from LN patients and healthy donors were used for IgG purification. A standard procedure using Protein G beads (GE Healthcare, Vélizy, France) was applied, following manufacturer’s recommendations. IgG protein concentration was measured by Nanodrop. The purity of IgG was determined by SDS-PAGE electrophoresis on 10% *v*/*v* precast polyacrylamide gels (Invitrogen and Life technologies, ThermoFisher Scientific, Illkirch-Graffenstaden, France).

### 4.4. Characterization of Autoantigen-Autoantibody Interaction by Surface Plasmon Resonance (SPR)

The interaction of the purified IgG with C1q, C1r and C1s was examined in real time using a ProteOn XPR36 SPR equipment (BioRad, Marne-la-Coquette, France) and BiaCore2000 (GE Healthcare, Vélizy, France). The covalent immobilization of C1q, C1r and C1s to a GLC sensor chip (BioRad) was performed following the manufacturer’s recommendations. Alternatively, CM5 chips for BiaCore were used. Protein G-purified IgG from patients with LN or healthy controls were injected for 300 s at six concentrations (300, 150, 75, 35, 17.5 and 0 µg/mL) diluted in PBS 0.005% Tween or 10 mM HEPES, 145 mM NaCl, 0.005% Tween 20 buffers. The dissociation was followed for 300 s. Bound protein was regenerated using 1 M NaCl, 50 mM NaOH regeneration buffer.

### 4.5. Functional Test for C1 Complex Formation

ELISA plates (Greiner Bio-One, Kremsmünster, Austria) were coated with a polyclonal rabbit anti-human C1q antibody (1:1000) and blocked with 1% *w*/*v* BSA. Next, IgG (200, 100, 50, 25, 12.5, 6.25, 3.13, 1.56 and 0 µg/mL), isolated from patients positive for anti-C1r, anti-C1s or anti-C1q antibodies, as well as from healthy controls were incubated with 10 µg/mL C1r/C1s and 2 µg/mL C1q in TTBS (a mixture of tris-buffered saline, TBS, and Tween 20), 10 mM CaCl_2_. After washing, the mixture of C1q, C1r, C1s and IgG was added to the wells. Bound C1 complex was revealed by a goat anti-human C1s anti-serum (1:500; Quidel, Eurobio Scientific, Les Ulis, France), followed by a rabbit anti-goat IgG-HRP conjugate (1:2000). The interactions were detected using 3,3′,5,5′-Tetramethylbenzidine (TMB) substrate at 450 nm. The test was adapted from assays developed in our previous research [5,68], where the reagents were shown to be functional and the C1 complex assembly in the fluid phase was validated.

### 4.6. Statistics Analysis

Statistical analysis was carried out using software GraphPad Prism 6.01. Mann–Whitney U test for continuous variables for 2-group comparisons was used. For comparison between more than two groups, a non-parametric test for independent samples, the Kruskal–Wallis one-way ANOVA, test was used. Fisher’s exact test was also performed for data analysis. Odds ratio (OR) and 95% confidence interval (CI) were calculated. The Spearman correlation was used to analyze the relativeness of the study parameters. Statistical significance was considered at *p* < 0.05.

## 5. Conclusions

The three components of the C1 complex act as autoantigens in LN; nevertheless, in this cohort, clinical relevance was observed only for anti-C1q. Therefore, among the C1 complex proteins, C1q remains the major autoantigen related to the classical pathway initiation to be screened in the clinical practice, correlating with disease activity and predicting flares.

## Figures and Tables

**Figure 1 ijms-23-09281-f001:**
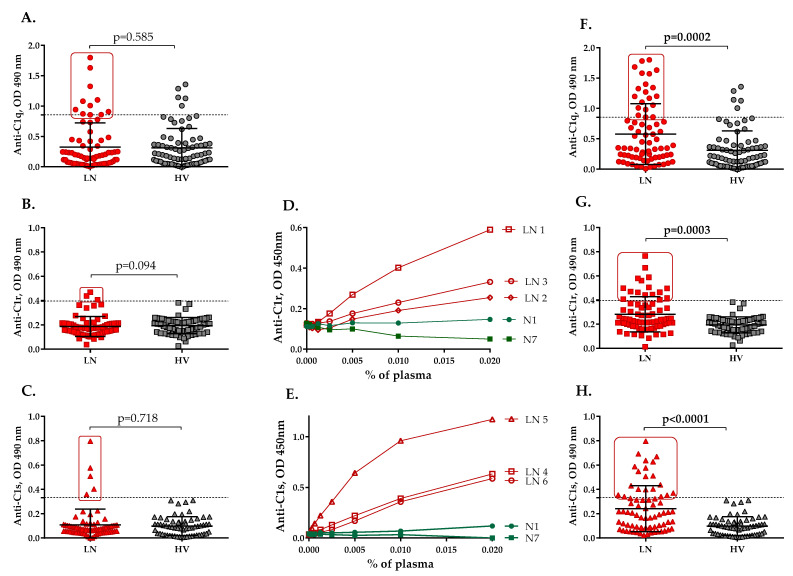
Presence of anti-C1 autoantibodies in LN. Reactivity against C1q (**A**), C1r (**B**) and C1s (**C**) from the plasma of patients with LN (LN) at the time of the first sampling, compared with healthy volunteers (HV), measured by ELISA. Reactivity against C1q (**F**), C1r (**G**) and C1s (**H**) from the plasma of patients with LN (samples with the highest levels of anti-C1 antibodies) (LN), compared with healthy volunteers (HV), measured by ELISA. The dotted line represents the positivity cut-off, determined as average ± 3 SD of the signal, obtained from the plasma of 81 healthy donors. N—normal control IgG, purified from healthy donors. Dose dependences of the IgG binding from different plasma samples to immobilized C1r (**D**) and C1s (**E**) were measured by ELISA.

**Figure 2 ijms-23-09281-f002:**
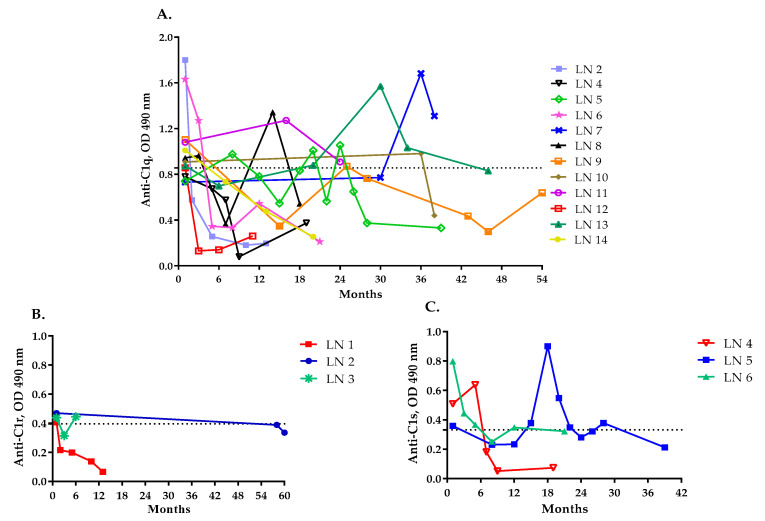
Dynamics of the levels of anti-C1 autoantibodies over time. Dynamics of the levels of anti−C1q (**A**), anti−C1r (**B**) and anti−C1s autoantibodies (**C**) over time. Only the patients for whom two or more samples were available are included in the figure. The dotted line represents the positivity cut-off, determined as average ± 3 SD of the signal, obtained from the plasma of 81 healthy donors.

**Figure 3 ijms-23-09281-f003:**
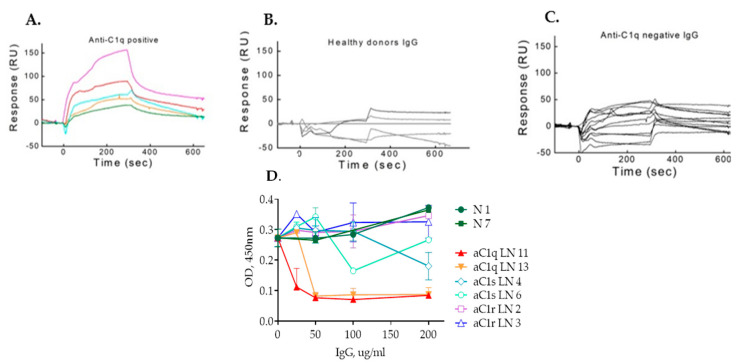
Assessment of the target antigen recognition by purified IgG. SPR sensorgrams for IgG binding to C1q of patients positive for anti−C1q antibodies by ELISA (**A**), of healthy volunteers (**B**) and of patients negative for anti-C1q antibodies by ELISA (**C**). Functional test for C1 complex formation (**D**).

**Figure 4 ijms-23-09281-f004:**
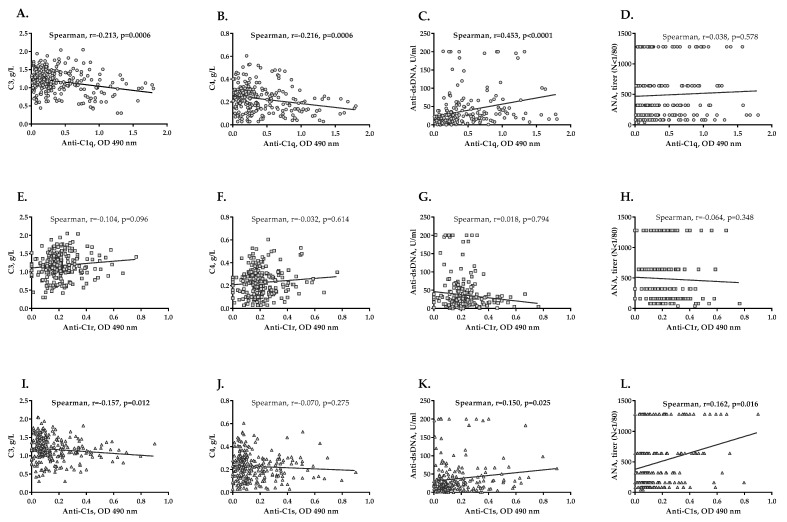
Correlation analysis between anti-C1q, anti−C1r and anti−C1s antibodies and markers of immunological activity in LN. Correlation between anti−C1q antibodies and levels of C3 (**A**), C4 (**B**), anti−dsDNA (**C**) and ANA titers (**D**). Correlation between anti−C1r antibodies and levels of C3 (**E**), C4 (**F**), anti−dsDNA (**G**) and ANA titers (**H**). Correlation between anti−C1s antibodies and levels of C3 (**I**), C4 (**J**), anti−dsDNA (**K**) and ANA titers (**L**).

**Figure 5 ijms-23-09281-f005:**
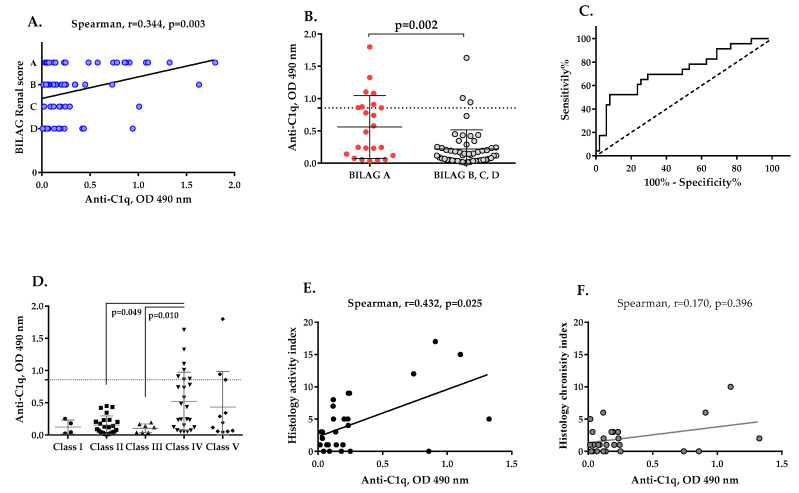
Association of the presence of anti-C1q antibodies in patients at the time of first sampling with the categories of the BILAG Renal score and histological signs of LN. Correlation between anti-C1q antibodies and categories of LN activity according to the BILAG Renal score (**A**). Comparative analysis between levels of anti-C1q antibodies in category A LN patients and patients classified into other categories according to the BILAG Renal score (**B**). ROC curve for anti-C1q antibodies in LN patients in BILAG category A towards LN patients in the rest of BILAG categories (**C**). Association between anti-C1q antibodies and histology class of LN (**D**). Correlation analysis between levels of anti-C1q antibodies and histological activity index (**E**) and histological chronicity index (**F**).

**Table 1 ijms-23-09281-t001:** Comparative analysis between levels of anti-C1q, anti-C1r and anti-C1s in the groups with and without histological signs of LN activity and chronicity.

Histologic Features *	Anti-C1q Levels, OD 490 nm Median (from–to)			Anti-C1r Levels, OD 490 nm Median (from–to)			Anti-C1s Levels, OD 490 nm Median (from–to)		
	Presence	Absence	*p*-Value	Presence	Absence	*p*-Value	Presence	Absence	*p*-Value
**Endocapillary proliferation**	0.139 (0.015–1.325)	0.180 (0.043–0.858)	0.846	0.165 (0.013–0.470)	0.180 (0.114–0.198)	0.698	0.071 (0.042–0.359)	0.092 (0.043–0.123)	0.194
**“Wire loop” deposits**	0.740 (0.12–1.325)	0.127 (0.015–1.103)	**0.019**	0.116 (0.084–0.47)	0.174 (0.013–0.438)	0.417	0.070 (0.054–0.359	0.073 (0.042–0.175)	0.975
**Fibrinoid necrosis/karyorrhexis**	0.235 (0.024–1.103)	0.136 (0.015–1.325)	0.269	0.174 (0.103–0.254)	0.148 (0.013–0.47)	0.581	0.072 (0.055–0.359)	0.074 (0.042–0.123)	0.616
**Cellular crescents**	0.909 (0.116–1.325)	0.128 (0.015–0.858)	**0.012**	0.116 (0.084–0.182)	0.171 (0.013–0.47)	0.248	0.061 (0.054–0.359)	0.076 (0.042–0.175)	0.640
**Interstitial inflammation**	0.198 (0.116–1.103)	0.089 (0.015–1.325)	0.167	0.175 (0.013–0.470)	0.159 (0.084–0.438)	0.744	0.070 (0.052–0.111)	0.085 (0.042–0.359)	0.303
**Glomerular sclerosis**	0.180 (0.015–1.325)	0.128 (0.024–0.858)	1.000	0.177 (0.084–0.215)	0.154 (0.013–0.470)	0.884	0.071 (0.043–0.175)	0.077 (0.042–0.359)	0.407
**Fibrous crescents**	0.909 (0.116–1.325)	0.128 (0.015–0.858)	**0.037**	0.179 (0.084–0.198)	0.165 (0.013–0.470)	0.731	0.061 (0.054–0.111)	0.076 (0.042–0.359)	0.473
**Tubular atrophy**	0.187 (0.016–1.103)	0.136 (0.015–1.325)	0.880	0.181 (0.103–0.438)	0.148 (0.013–0.470)	0.228	0.066 (0.042–0.175)	0.077 (0.047–0.359)	0.280
**Interstitial fibrosis**	0.231 (0.016–1.103)	0.127 (0.015–1.325)	0.382	0.171 (0.103–0.47)	0.165 (0.013–0.438)	0.662	0.071 (0.043–0.175)	0.081 (0.042–0.359)	0.396

* Only patients where the time between blood sampling and kidney biopsy is up to 12 months were selected.

## Data Availability

The data presented in this study are available upon request from the corresponding author at e-mail: maria.radanova@gmail.com. The data are not publicly available due to their containing information that could compromise the privacy of the research participants.

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
