# Peer review of "Autoantibodies against Complement Classical Pathway Components C1q, C1r, C1s and C1-Inh in Patients with Lupus Nephritis"

_ijms, 2022, doi:10.3390/ijms23169281_

Round 1

Reviewer 1 Report

This paper describes the occurrence of autoantibodies targeting the components of the complement C1 complex, i.e. C1q, C1r and C1s, as well as C1 Inhibitor in patients with SLE. While no autoantibodies against C1 Inh were found, autoantibodies against C1q were frequently observed and correlated with disease activity/manifestations. In addition, autoantibodies against C1r and C1s were also found but did not correlate with disease manifestations. While the data on anti-C1r/s are novel, at least parts of the observations on anti-C1q have already been described by previous studies on these autoantibodies. Additional data are primarily on analyzing anti-C1q by SPR.

Comments

The authors describe some discrepancy in results between antibody detection by ELISA versus SPR. This discrepancy should be discussed. Any explanation ?

One major finding is the description of anti-C1r/s antibodies. Were these antigens also recognized when using C1 complexes as the antigen ? Of particular interest in samples that were negative for anti-C1q.

In paragraph 2.3 a ‘functional’ assay is described in which the assembly of C1 complexes in the presence of anti-C1 antibodies was studied. As the assembly of the C1 complex was performed in the fluid phase: Was the assembled C1 complex functional, i.e. able to activate complement (e.g. C4 cleavage), and if yes, was this functionality reduced in the presence of anti-C1r/s as well ? Did the authors make sure, that the SLE patient derived IgG preparations were free of antibodies against the capturing rabbit antibody that then could have been detected in the sense of cross-reaction by the developing antibody (i.e. negative control with absence of the complement components) ?

It is not getting clear at which time the renal biopsies had been performed in relation to the analysed antibody titer.

What were the positive controls used for autoantibodies against C1r, C1s and C1 Inh ?

Minor:

Introduction, second line: LN is not necessarily the most severe manifestation of SLE (e.g. think of ZNS involvement). Please correct.

Results, first line: Please stay consistent by the use of anti-C1q, anti-C1r, anti-C1s. Same in the title and first line of results part 2.4. as well as in title of results 2.5

Results, last sentence of paragraph 2.1.: What did the authors mean with “showed active variability over time” ?

Overall, I have the impression that most antibody titers were rather low/more frequently negative than expected (incl. data on ANA, anti-dsDNA). Could this be due to storage conditions (time, temperature, repetitive use) ?

Author Response

Reviewer 1

This paper describes the occurrence of autoantibodies targeting the components of the complement C1 complex, i.e. C1q, C1r and C1s, as well as C1 Inhibitor in patients with SLE. While no autoantibodies against C1 Inh were found, autoantibodies against C1q were frequently observed and correlated with disease activity/manifestations. In addition, autoantibodies against C1r and C1s were also found but did not correlate with disease manifestations. While the data on anti-C1r/s are novel, at least parts of the observations on anti-C1q have already been described by previous studies on these autoantibodies. Additional data are primarily on analyzing anti-C1q by SPR.

Comment:

The authors describe some discrepancy in results between antibody detection by ELISA versus SPR. This discrepancy should be discussed. Any explanation?

Author’s Reply:

We thank the reviewer for this question, which intrigues us a lot. One possible explanation for this phenomenon is the conformational change in C1q, needed for the binding of the anti-C1q Antibodies. Indeed, anti-C1q Abs do not interact with C1q in solution, the immunodominant epitope in the collagenous region has to be exposed upon immobilization on the plate or in tissues. This cryptic epitope may not be fully exposed upon C1q immobilization to the biosensor chip. On the contrary, other cryptic epitopes may be exposed, invisible by the routine ELISA technique. Also ELISA and SRP are quite different methods that are founded on different principles. SPR is a more sensitive method that can detect lower affinity antibodies having a faster rate of dissociation from the antigen. Through the SPR method, binding events are observed in real time without the long incubation steps characteristic of ELISA. It is known that low-affinity antibodies cannot be detected by ELISA (Wadhwa M, Knezevic I, Kang HN, Thorpe R. Immunogenicity assessment of biotherapeutic products: An overview of assays and their utility. Biologicals. 2015 Sep;43(5):298-306.). In the present study, the SPR method was mainly used to validate ELISA results showing only pathologically elevated antibodies against the respective complement antigen.

The following text was added in the Discussion section (form 264 to 275 lines):

The discrepancy between ELISA and SPR found for some samples could be related to the fact that anti-C1q bind weakly (if any) to soluble C1q or to C1q within the C1 [33-35] and a conformational change, induced by the tissue/ELISA plate is necessary to reveal the cryptic immunodominant epitope(s), such as the so called “A08” epitope [36, 37]. These cryptic epitopes may not be correctly expressed upon C1q immobilization to the biosensor chip and the weak titer anti-C1q may not be able to bind them in the cases when loss of reactivity was detected. On the contrary, other cryptic epitopes may be exposed upon SPR type of immobilization (which is covalent linking via the Lys residues to a flexible dextran matrix, creating a semi-solid, semi-fluid phase model) revealing novel, previously undetected ant-C1q antibodies, invisible by the standard ELISA method. These antibodies have to be further explored to evaluate their functional and clinical relevance.

Comment:

One major finding is the description of anti-C1r/s antibodies. Were these antigens also recognized when using C1 complexes as the antigen? Of particular interest in samples that were negative for anti-C1q.

Author’s Reply:

The use of C1 complex as an antigen is tricky as it can dissociate upon immobilization on the ELISA plate. We have not explored C1 as an antigen also because the pathogenic role of C1q was suggested to be limited to tissues or organs in which C1q is deposited and not associated with the serine proteases C1r and C1s (Uwatoko S, Gauthier VJ, Mannik M. Autoantibodies to the collagen-like region of C1Q deposit in glomeruli via C1Q in immune deposits. Clin Immunol Immunopathol. 1991 Nov;61(2 Pt 1):268-73.). Nevertheless, we agree with the reviewer that this is interesting and has to be tested.

We added some sentences in the manuscript discussion on this aspect (from 275 to 277 lines):

Moreover, immobilized C1 complex could be tested to reveal whether the anti-C1q, anti-C1r, anti-C1s or even the negative IgG samples could reveal some binding due to exposure of cryptic neoepitopes.

Comment:

In paragraph 2.3 a ‘functional’ assay is described in which the assembly of C1 complexes in the presence of anti-C1 antibodies was studied. As the assembly of the C1 complex was performed in the fluid phase: Was the assembled C1 complex functional, i.e. able to activate complement (e.g. C4 cleavage), and if yes, was this functionality reduced in the presence of anti-C1r/s as well?

Author’s Reply:

We thank the reviewer for these questions helping us to clarify the description of our assay. It is based on previously published experimental setting, in which we developed the assay to test the presence of circulating C1 complex in serum or the functional activity of C1r and C1s. The same reagents (C1q, C1r, C1s from CompTech) were tested in a complement-activating assay for cleavage and deposition of C4b and cleavage of C2 and are validated as functional (Roumenina LT, Daugan MV, Noé R, Petitprez F, Vano YA, Sanchez-Salas R, et al. Tumor Cells Hijack Macrophage-Produced Complement C1q to Promote Tumor Growth. Cancer Immunol Res. 2019 Jul;7(7):1091-1105.). The complex was formed in the fluid phase as here. We have not tested the functionality here and we agree that this test is important. Nevertheless, the immobilization of C1q on the surface coated with the rabbit anti-C1q antibody was apparently efficient to reveal the cryptic epitopes necessary for the recognition of C1q by the anti-C1q antibodies from the patients. This resulted in decrease in the C1 complex formation, supporting historical findings that anti-C1q bind to C1q and not the C1 complex (Uwatoko S, Gauthier VJ, Mannik M. Autoantibodies to the collagen-like region of C1Q deposit in glomeruli via C1Q in immune deposits. Clin Immunol Immunopathol. 1991 Nov;61(2 Pt 1):268-73.) This is in line with recent findings that the functions of free C1q, such as the clearance of the apoptotic cells, are affected by the anti-C1q auto-antibodies.

We have added a comment on this in the Materials and Methods section (from 436 to 438 lines):

The test is adapted from assays developed in our previous research [68, 69], where the reagents were shown to be functional and the C1 complex assembly in fluid phase was validated.

And in the Discussion section (from 277 to 285 lines):

Nevertheless, previous finding and our results here suggest that the anti-C1q bind preferentially to the free C1q [38] and even can prevent/dissociate the C1 complex formation. This is in line with the findings that anti-C1q perturb the functioning of free C1q in the clearance of apoptotic cells [39] and trigger a pro-inflammatory phenotype in macrophages, reversing the anti-inflammatory effects of immobilized C1q alone [40]. Moreover, C1q/anti-C1q-primed monocytes induce pathological T cell activation via direct CD40-CD154 interaction [41]. Altogether, these results suggest that free and not complexed C1q is the main target of the anti-C1q autoantibodies in LN.

And in the Discussion section (from 341 to 343 lines):

Further studies are needed to explore the potential functional relevance of the anti-C1s as potentiators of the C4 and/or C2 cleavage and factors affecting the non-canonical functions of C1s, independent of the C1 complex.

Comment:

Did the authors make sure, that the SLE patient derived IgG preparations were free of antibodies against the capturing rabbit antibody that then could have been detected in the sense of cross-reaction by the developing antibody (i.e. negative control with absence of the complement components)?

Author’s Reply:

It is highly unlikely that such non-specific interaction could have occurred in our assay. If there were antibodies cross reacting with the developing antibody, we should have registered aberrantly strong signal in the detection. It was not the case. On the contrary, the anti-C1q positive IgG caused reduced detection of the C1 complex.

Comment:

It is not getting clear at which time the renal biopsies had been performed in relation to the analysed antibody titer.

Author’s Reply:

We thank the reviewer for this very important comment. Indeed, we used only renal biopsies performed at the time of sampling for autoantibodies or performed less than 1 year from the testing for autoantibodies. Samples for which the biopsy was older than 1 year were excluded from the analysis of the correlation with the histological findings.

This was indicated in the Material and Methods section (from 371 to 373 lines):

Only biopsies performed less than 12 months year from the sampling for autoantibodies were included in the analysis for correlation with the histological findings.

Comment:

What were the positive controls used for autoantibodies against C1r, C1s and C1 Inh?

Author’s Reply:

As these assays are not standardized for clinical diagnostics, we did not have a positive control, i.e. a known patient with high titers. We counted on our cohort to find positive samples to be used in future assays to control for the binding. We agree with the reviewer that in the future a standard has to be created to calculate these antibodies in arbitrary units comparable between different laboratories. This will become relevant if such antibodies are found to be of interest for the clinical practice in other diseases or other forms of SLE (without LN for example, which has to be tested, as our cohort was exclusively composed of patients with LN).

Minor Comment:

Introduction, second line: LN is not necessarily the most severe manifestation of SLE (e.g. think of ZNS involvement). Please correct.

Author’s Reply:

Thanks for this precision, we wrote “one of its common and severe manifestations is LN” (40 line).

Minor Comment:

Results, first line: Please stay consistent by the use of anti-C1q, anti-C1r, anti-C1s. Same in the title and first line of results part 2.4. as well as in title of results 2.5

Author’s Reply:

We have corrected this, thank you.

Minor Comment:

Results, last sentence of paragraph 2.1.: What did the authors mean with “showed active variability over time”?

Author’s Reply:

The word “active” is indeed not necessary. We modified to “showed variation over time” (113 line).

Minor Comment:

Overall, I have the impression that most antibody titers were rather low/more frequently negative than expected (incl. data on ANA, anti-dsDNA). Could this be due to storage conditions (time, temperature, repetitive use)?

Author’s Reply:

The levels of ANA and anti-dsDNA are measured in the course of routine clinical examinations and were performed in the certified diagnostic laboratory of the ISUL hospital when the samples were taken by a routine protocol. We used only “leftover” samples for our analyses, but they were aliquoted at the moment of sampling and stored at -20°C until transport to the research lab and there – at -80°C. Therefore, the titres of ANA and anti-dsDNA could not be affected by the timing, storage or repetitive use.

Moreover, in Materials and Methods section we described the LN cohort and presented the clinical laboratory and immunological parameters of patients at the moment of the first sampling, which was not necessary during an acute phase. These patients are with different duration of disease and therapy. We re-analyzed our dataset and took as a cross-section not only the first available sample, but the sample with the highest titers of anti-C1q autoantibodies. When we present the cohort by levels of anti-C1q using the sample with the highest level of anti-C1q for each patient the number of positive for ANA and anti-dsDNA antibodies rises to 76.67% and to 50.82%, respectively.

We are grateful to this reviewer for the constructive criticism and valuable advices, which we have taken into account to improve the manuscript.

Reviewer 2 Report

Here, this study investigated the autoantibodies against complement classical pathway components C1q, C1r, C1s and C1-Inh in patients with lupus nephritis, and the author found that the three components of the C1 complex act as autoantigens in LN nevertheless that in this cohort clinical relevance was observed only for anti-C1q.

Questions:

1. Is OD.490nm the standard unit for ELISA in this study?

2. Why chose two methods to detect the autoantibodies ? as stated in the manuscript (Adiscrepancy was observed between the two methods. Some low titer anti-C1q IgG, detected by ELISA, were negative in SPR analysis and vice versa, (data not shown).

3.For Classe IV group, what about the difference between positive and negative for anti-C1q antibodies? 

4. Why just chose ±3SD, not ±2SD?  

Author Response

Reviewer 2

Here, this study investigated the autoantibodies against complement classical pathway components C1q, C1r, C1s and C1-Inh in patients with lupus nephritis, and the author found that the three components of the C1 complex act as autoantigens in LN nevertheless that in this cohort clinical relevance was observed only for anti-C1q.

Comment:

  1. Is OD.490nm the standard unit for ELISA in this study?

Author’s Reply:

For the auto-antibodies detection we used OPD as a substrate and with this system the signal is revealed as optical density at 490 nm.

To clarify this point, the following sentence was added in the Material and Methods section (from 402 to 403 lines):

… and the signal was detected as optical density (OD) at 490nm.

Comment:

  1. Why chose two methods to detect the autoantibodies? as stated in the manuscript (Adiscrepancy was observed between the two methods. Some low titer anti-C1q IgG, detected by ELISA, were negative in SPR analysis and vice versa, (data not shown).

Author’s Reply:

We are grateful to the reviewer for this question. It is raised also by the other reviewer. We used the SPR as a validation method and to evaluate the association and dissociation rate of the antibodies. For most of the anti-C1q+ samples we obtained good agreement between ELISA and SPR, nevertheless, some samples showed discrepancy. One possible explanation for this phenomenon is the conformational change in C1q, needed for the binding of the anti-C1q Antibodies. Indeed, anti-C1q Abs do not interact with C1q in solution, the immunodominant epitope in the collagenous region has to be exposed upon immobilization on the plate or in tissues. This cryptic epitope may not be fully exposed upon C1q immobilization to the biosensor chip. On the contrary, other cryptic epitopes may be exposed, invisible by the routine ELISA technique.

The following text was added in the Discussion section (from 264 to 275 lines):

The discrepancy between ELISA and SPR found for some samples could be related to the fact that anti-C1q bind weakly (if any) to soluble C1q or to C1q within the C1 [33-35] and a conformational change, induced by the tissue/ELISA plate is necessary to reveal the cryptic immunodominant epitope(s), such as the so called “A08” epitope [36, 37]. These cryptic epitopes may not be correctly expressed upon C1q immobilization to the biosensor chip and the weak titer anti-C1q may not be able to bind them in the cases when loss of reactivity was detected. On the contrary, other cryptic epitopes may be exposed upon SPR type of immobilization (which is covalent linking via the Lys residues to a flexible dextran matrix, creating a semi-solid, semi-fluid phase model) revealing novel, previously undetected ant-C1q antibodies, invisible by the standard ELISA method. These antibodies have to be further explored to evaluate their functional and clinical relevance.

Comment:

  1. For Classe IV group, what about the difference between positive and negative for anti-C1q antibodies?

Author’s Reply:

We are grateful to the reviewer for this suggestion. We took the samples form the cohort taken from patients when they were in Class IV and compared the available parameters. A significant difference was detected for the immunological parameters between anti-C1q positive and anti-C1q negative class IV samples. Anti-C1q positive class IV patients had significantly elevated anti-dsDNA, reduced C3 and C4, a tendency towards elevated proteinuria and significantly higher BILAG class compared to the ones with class IV but negative for anti-C1q. This information was given in the results section.

The following text was added in the Results section (from 215 to 221 lines):

When the samples of patients with Class IV only were stratified according to their anti-C1q status, the anti-C1q positive samples had significantly elevated anti-dsDNA (p=0.0128), lower C3 (p=0.0057) and C4 (0.02), and a tendency to elevated proteinuria (p=0.0621). Moreover, anti-C1q positive Class IV patients had more frequently A class according to BILAG Renal score, country to the anti-C1q negative ones (p=0.0286). No difference was found for the eGFR, the index of activity, the urea sediment, ANA, the age or the frequency of anti-C1r or anti-C1s.

Comment:

  1. Why just chose ±3SD, not ±2SD?

Author’s Reply:

We used ±3SD instead of ±2SD to define sure seropositive patients. Of note, certain plasma samples from healthy volunteers were anti-C1q positive, in agreement with other studies (Mahler M, van Schaarenburg RA, Trouw LA. Anti-C1q autoantibodies, novel tests, and clinical consequences. Front Immunol. 2013 May 14;4:117; Radanova M, Vasilev V, Deliyska B, Kishore U, Ikonomov V, Ivanova D. Anti-C1q autoantibodies specific against the globular domain of the C1qB-chain from patient with lupus nephritis inhibit C1q binding to IgG and CRP. Immunobiology. 2012 Jul;217(7):684-91.). These outliers were excluded from the determination of the cut-off level but are presented in the figures.

We are grateful for the questions and suggestions of this reviewer. We are hoping that we have understood your comments and our answers are acceptable.
